# Palaeoclimate significance of speleothems in crystalline rocks: a test case from the Lateglacial and Early Holocene (Vinschgau, northern Italy)

Gabriella Koltai[1*], Hai Cheng[2], Christoph Spötl[1]

[1] Institute of Geology, University of Innsbruck, Innrain 52, 6020 Innsbruck, Austria
[2] Xi'an Jiaotong University, Institute of Global Environmental Change, 28 Xianning West Road, Xi`an 710049, Shaanxi, China

*Correspondence to*: gabriella.koltai@uibk.ac.at

**Abstract:** Partly coeval flowstones formed in fractured gneiss and schist were studied to test the palaeoclimate significance of this new type of speleothem archive on a decadal to millennial timescale. The samples encompass a few hundred to a few thousand years of the Lateglacial and the Early Holocene. The speleothem fabric is primarily comprised of columnar fascicular optic calcite and acicular aragonite, both being indicative of elevated Mg/Ca ratios in the groundwater. Stable isotopes suggest that aragonite is more prone to disequilibrium isotope fractionation driven by evaporation and prior calcite/aragonite precipitation than calcite. Changes in mineralogy are therefore attributed to these two fracture-internal processes rather than to palaeoclimate. Flowstones formed in the same fracture show similar $\delta^{18}O$ changes on centennial scales, which broadly correspond to regional lacustrine $\delta^{18}O$ records, suggesting that such speleothems may provide an opportunity to investigate past climate conditions in non-karstic areas. The shortness of overlapping periods in flowstone growth and the complexity of in-aquifer processes, however, render the establishment of a robust stacked $\delta^{18}O$ record challenging.

## 1 Introduction

Speleothems from karst caves have contributed important information on past climate change from orbital to seasonal time scales worldwide (e.g. Johnson et al., 2006; Boch et al., 2011; Fohlmeister et al., 2012; Wang et al., 2014; Webb et al., 2014; Luetscher et al., 2015; Cheng et al., 2016). While the importance of these speleothems as palaeoclimate archives is firmly established, very little is known about the palaeoclimate potential of such deposits from non-carbonate settings. Since only ~ 20% of the ice-free continental area of the Earth consists of carbonate rocks prone to karstification (Ford and Williams, 2007) speleothems from non-karstic settings may provide relevant palaeoclimate archives for semi-arid (Koltai et al., 2017) to super-humid (Schmipf et al., 2011) to subglacial environments (Frisia et al., 2017).

In comparison to other palaeoclimate archives, a key advantage of speleothems is that they can be dated with high precision by U-Th techniques up to about half a million years (e.g. Richards and Dorale, 2003; Scholz and Hoffmann, 2008; Cheng et al., 2013). Since silicate rocks usually contain much more uranium than limestones or dolostones (Wedepohl, 1995), the $^{238}U$ content of speleothems forming in cavities and fractures of crystalline rocks is commonly orders of magnitude higher, and therefore require very little sample amount to yield high resolution chronologies (Spötl et al., 2002; Koltai et al., 2017).

The most widely used proxies of karst speleothems include stable oxygen and carbon isotopes, trace elements, growth rate, and fabric changes (e.g. Frisia et al., 2003; McDermott, 2004; McMillan et al., 2005; Wassenburg et

al., 2012). Although the majority of speleothem-based climate reconstructions utilized calcite speleothems, several studies indicate that aragonite can also record past climate, in particular rainfall variability (e.g. Polyak & Asmerom, 2001; Ridley et al., 2015; Wassenburg et al., 2016).

Here, we present a well-dated, multi-proxy record of eight fast growing flowstones from a non-karstic setting in a dry inner-alpine setting (Vinschgau). Unlike speleothems formed in karst caves, these calcite and aragonite flowstones were deposited in near-surface fractures created by gravitational mass movements. The aim of this study was to test the reliability of climate proxies preserved in this hitherto largely neglected type of speleothem archive. To this end we chose the Lateglacial to Early Holocene time interval which is among the best characterised periods in the late Quaternary of the Alps based on studies of lake sediments (e.g., von Grafenstein et al., 1999, 2013; Magny et al., 2001; Ilyashuk et al., 2009; Lauterbach et al., 2011; Heiri et al., 2014), palaeoglaciers (e.g., Kerschner et al., 2000; Ivy-Ochs et al., 2008; Kerschner and Ivy-Ochs, 2008) and speleothems (Wurth et al., 2004; Luetscher et al., in prep.). By comparing our speleothem data to these published records we explore the extent to which these non-karstic speleothems register the well-documented succession of rapid climate change during this time period.

## 2 Site description and samples

### 2.1 Study site

The Vinschgau is a E-W trending inner-alpine valley shielded by high mountain chains in the north, west and south. Although it is one of the driest valleys of the Eastern Alps today (Fliri, 1975; ZAMG, 2015), local climate archives such as large debris-flow fans (which are largely inactive today) point to periods of distinctly more humid climate during the Lateglacial and Holocene in this valley.

The study area is comprised of paragneiss, orthogneiss and schists that are heavily fractured as a result of deep-seated gravitational mass movements along the Sonnenberg (Ostermann et al., submitted). These fractures provide pathways for groundwater flow. Due to pronounced water-rock interaction dominated by pyrite oxidation and the presence of large internal mineral surfaces created by the mass movements, these waters are highly mineralized. Their electric conductivity varies between 730 and 2300 µS/cm and they are characterised by elevated molar Mg/Ca ratios between 0.3 and 3.3. Due to evaporation and concomitant $CO_2$ degassing groundwater reaches supersaturation on the south-facing Sonnenberg slope, where springs are supersaturated with respect to calcite and also to aragonite except for one. Speleothems form as calcite and aragonite flowstones in the shallow subsurface and calcitic freshwater tufa deposits are present on the surface (Spötl et al., 2002; Koltai et al., 2017).

### 2.2 Speleothem samples

Eight vein-filling flowstones (Suppl. Fig. 1) were obtained from three different fractures (Fig. 1). It must be emphasized that the samples were found in the debris next to the fractures and their exact position within the fractures is unknown. LAS 1, LAS 2 and LAS 21 were collected at Törgltal (TR, 46.631° N, 10.680° E), while LAS 6, LAS 34 and LAS 72 are from Stollenquelle (SQ, 46.630° N, 10.683° E). These two sites are less than 1 km apart. The third site (Kortsch-KO, 46. 635° N, 10.763° E) is situated approximately 6 km east of SQ. Two samples (LAS 10 and 19) were collected there.

Aragonite is present near the top of samples LAS 2, LAS 6 and LAS 21. Thus, the uppermost 12, 15 and 6 mm
of LAS 2, LAS 6 and LAS 21, respectively, were not included in this study. As LAS 10 grew between 9.26 ±
0.10 ka and 10.22 ± 0.06 ka according to the depth-age model, the present study focuses only on the lower 21.4
mm of this flowstone (Suppl. Fig. 1).
**3 Methods**
**3.1 Petrography**
Thin sections were analysed under transmitted-light and blue-light epifluorescence microscopy in order to
identify characteristic fabrics and areas of replacement of aragonite by calcite. Furthermore, small aliquots of
carbonate powder were obtained from LAS 2, LAS 6 and LAS 34 using a handheld dental drill in order to
determine the mineralogical composition by X-ray diffractometry (XRD).
**3.2 Stable isotopes**
Samples for stable oxygen and carbon isotope analyses were micromilled at different resolutions. LAS 2 was
sampled at 0.1 mm intervals, LAS1, LAS 6, LAS 34 and LAS 72 were micromilled at 0.15 mm increments. In
order to reach a multi-annual resolution (3 years) LAS 6 was also analysed by using three 2.5 mm-long parallel
tracks milled with a 0.8 mm offset perpendicular to the lamination. LAS 10 and LAS 19 were analysed at 0.2
mm, while LAS 21 was analysed at 0.25 mm resolution. Stable isotope measurements were performed using a
Thermo Fisher Scientific DELTA$^{plus}$XL mass spectrometer. Isotope values are reported against the VPDB scale
and the long-term analytical precision (1σ) of the $\delta^{18}O$ and $\delta^{13}C$ measurements is 0.08 and 0.06‰, respectively
(Spötl, 2011).
**3.3 U-Th dating**
A total of 77 powder samples were prepared for radiometric dating. If present, primary aragonite was preferred
over calcite, since aragonite-to-calcite transformation may alter the geochemical composition (e.g. Lachniet et
al., 2012; Domínguez-Villar et al., 2016). U-Th dates were divided amongst samples as follows: eleven dates
measured from LAS 2 and LAS 6, ten from LAS 1, eight from LAS 10 and LAS 21, fourteen from LAS 19, nine
from LAS 72, and six from LAS 34. Aliquots were obtained for U-Th dating from distinct growth layers using a
handheld drill. The weight of individual subsamples ranged between 1.0 and 9.0 mg.
The samples were analysed at the Xi`an Jiaotong University (China) following standard chemistry procedures of
Edwards et al. (1987) to separate uranium and thorium. U and Th isotopes were analysed individually by using a
multicollector inductively coupled plasma mass spectrometer (Thermo Fischer Neptune Plus) as described by
Shen et al. (2012) and Cheng et al. (2013). Final $^{230}$Th ages are given with their 2σ uncertainties as years before
1950 AD (BP). Corrected and uncorrected results are given in Suppl. Table 1. Corrected ages assume an initial
$^{230}$Th/$^{232}$Th ratio of 4.4 ± 2.2 x 10$^{-6}$ of bulk Earth (Wedepohl, 1995). Separate age models for all samples were
built using the StalAge algorithm (Scholz and Hoffman, 2011).

## 4 Results

### 4.1 Petrography

The crystal fabric of LAS 6, LAS 10 and LAS 19 is dominated by columnar fascicular optic calcite (Cfo; Frisia, 2015), showing undulose extinction due to the systematic change in the orientation of the c-axes (Kendall, 1985; Richter et al., 2011; Frisia, 2015). Detritus-rich layers are locally present in LAS 10 and LAS 19. Their average thickness varies from 25 to 50 μm in LAS 10 and between 50 and 75 μm in LAS 19, but in the latter sample detritus-rich layers up to 0.25 mm thick are also present.

LAS 72 is comprised of white acicular aragonite, while translucent calcite (Cfo) is also present in LAS 1, 2, 21 and 34 (Fig. 2a). XRD results indicate 100% calcite in the calcite layers of LAS 2 and 34. In LAS 2 and also in LAS 21 pristine aragonite islands are locally present in the calcite fabric that shows no sign of dissolution suggesting co-precipitation of aragonite and calcite (Fig. 2b).

Thin-section analyses of LAS 34 revealed the presence of mosaic calcite (Fig. 2c), a fabric indicative of recrystallisation (e.g. Frisia, 2015). As recrystallization may have modified the geochemical composition of the calcite, only the aragonite fabric is discussed further in this study. None of the other samples shows any sign of diagenetic alteration.

From the 60 until 110 mm dft LAS 6 exhibits annual calcite lamina couplets which can be observed macroscopically as successive white and translucent laminae. The white laminae are rich in opaque particles, whose organic origin is confirmed by their strong epifluorescence (Koltai et al., 2017). Similarly, the inclusion-rich layers in the fascicular optic calcite of LAS 10 and LAS 19 show excitation under epifluorescence. Furthermore, weakly fluorescent laminae are present in LAS 72, while both calcite and aragonite layers in LAS 2 and LAS 21 appear dull.

### 4.2 Stable isotope composition

The summarized results of the stable isotope analyses are presented in Table 1. High-resolution stable isotope profiles of the flowstones collected near TR show very similar values for $\delta^{18}O$, while carbon isotope values are more diverse. Even though $\delta^{13}C$ minima are identical within the $1\sigma$ analytical error in these samples, the highest carbon isotope values range from 2.4 to 7.3 ‰ (Table 1).

As Table 1 shows, the SQ samples are characterised by different values regarding both $\delta^{18}O$ and $\delta^{13}C$. The two flowstones dominated by aragonite (LAS 34 and LAS 72) exhibit higher isotope values, while the calcite samples from KO are characterised by the lowest $\delta^{13}C$ values.

The majority of the flowstones do not exhibit a correlation ($R^2<0.60$) between $\delta^{13}C$ and $\delta^{18}O$ values, with the exception of three samples (LAS 2, LAS 34 and LAS 72) which show a significant correlation between the two isotopes with slopes of regression varying from 2.5 to 3.4 (Fig. 3). The crystal fabric of these samples is dominated by acicular aragonite. In LAS 2 the covariance of $\delta^{13}C$ and $\delta^{18}O$ is characterised by almost identical slopes of the regression lines for both aragonite ($\delta^{13}C= 2.7*\delta^{18}O + 32.4$, $R^2=0.79$) and calcite ($\delta^{13}C= 2.9*\delta^{18}O + 35.2$, $R^2=0.60$) (Fig. 3A). On the contrary, LAS 1 does not show any covariance for calcite ($R^2= 0.33$) and aragonite ($R^2=0.13$). The number of stable isotopes analyses (n=7) in the aragonite of LAS 21 was too small to investigate the relationship of $\delta^{13}C$ and $\delta^{18}O$ variability.

Carbon isotope values mostly follow the first-order changes of oxygen isotopes in all samples except LAS 1. However, the relationship between the two isotopes may vary within a given sample. In LAS 6 and LAS 21 this

relationship breaks down in the topmost 13 and 7 mm distance from top (dft), respectively, while in LAS 19
rising $\delta^{18}O$ values correspond to decreasing $\delta^{13}C$ levels from 66 to 54 mm (dft).
Major changes in $\delta^{18}O$ values are observed in LAS 2 and LAS 19. The former sample exhibits generally high
oxygen isotope values in the aragonite growth phase from the bottom until 50 mm (dft), interrupted by periods of
lower $\delta^{18}O$ values. A 0.8 ‰ decrease is seen between 74 and 72 mm (dft) coinciding with the presence of a
calcite layer. A gradual shift of 2.2 ‰ towards lower values is observed from 52 to 47 mm (dft) and is
independent of the fabric, while a 1.6 ‰ rise in $\delta^{18}O$ characterises the calcite from 17 to 14 mm (dft). In LAS 19
a significant 3.2 ‰ shift towards more positive $\delta^{18}O$ values occurs from 66 to 54 mm (dft), followed by a
decrease in carbon isotope values. Moreover, in all samples $\delta^{18}O$ values show high-frequency changes of
different amplitude (0.5 to 1.5 ‰) while no major trend is observed.
**4.3 Chronology**
The Vinschgau samples show exceptionally high $^{238}U$ concentrations, ranging from ca. 1.5 to 1200 ppm (Table
1) and are among the most U-rich speleothems ever reported (Spötl et al., 2002; Kelly et al., 2003). $\delta^{234}U$ ranges
from 7 to 100 ‰. The $^{232}Th$ content is highly variable and fluctuates between 61 ppt and 178 ppb (Suppl. Table
1). Except for three subsamples of LAS 19 all samples show high $^{230}Th/^{232}Th$ activity ratios and thus, excess
$^{230}Th$ has no significant influence on the final ages.
Of the 77 total dates measured, 74 are in stratigraphic order within their 2σ uncertainties. The three dated offsets
(from samples L6-54, L6-79 and L1-38.5) are 5, 2 and 60 years beyond stratigraphic order, respectively (Suppl.
Table 1). As these differences represent less than 0.5, 0.25 and ~2‰ age deviation for L6-54, L6-79 and LAS 34,
respectively, we do not consider these ages as outliers and include them in the age models (Suppl. Figs. 1-2).
LAS 1 formed between 12.99 ± 0.05 and 12.01 ± 0.03 ka, while LAS 2 grew uninterruptedly between 14.18 ±
0.03 and 12.12 ± 0.03 ka. The last sample from the TR site, LAS 21 initiated deposition at 12.28 ± 0.03 and grew
continuously until 11.68 ± 0.02 ka BP (Suppl. Figs. 2-3).
LAS 6 from the SQ site formed between 12.06 ± 0.04 and 11.68 ± 0.03 ka. Similarly to LAS 2, growth of LAS
34 commenced 14.18 ± 0.03 ka and ended 12.54 ± 0.03 ka, showing no major growth interruptions. LAS 72
provides a record between 11.64 ± 0.04 and 10.03 ± 0.03 ka. The age model of LAS 72 shows that there may
have been a hiatus from 10.54 to 10.30 ka (modelled ages) corresponding to 13 to 14 mm on the depth scale
(Suppl. Fig. 3d). Yet, as thin-section analysis provided no evidence for a growth interruption (e.g. corrosion
layer) we attribute this to an interval of slow growth rates.
The U-Th dates of the studied section of LAS 10 range from 9.94 ± 0.03 to 10.21 ± 0.06 ka, while LAS 19
started to form in the YD at 11.98 ± 0.05 ka and stopped growing in the Early Holocene at 10.78 ± 0.04 ka
(Suppl. Figs. 2-3).
**5 Discussion**
**5.1 Stable isotope systematics**
**5.1.1 $\delta^{18}O$**
In karstic settings, speleothem $\delta^{18}O$ values depend on the $\delta^{18}O$ composition of drip water and the cave air
temperature, the latter influencing water-carbonate fractionation factors for both calcite and aragonite (e.g.
McDermott, 2004; Lachniet, 2015). Modern spring monitoring in the Vinschgau suggests that the speleothem-

forming waters are part of a larger groundwater system recharging at an elevation ranging from about 1200 to 2100 m a.s.l. Minimal variation in stable isotope composition and low tritium content point to long mean residence times of up to several decades (Spötl et al., 2002).

LAS 6 exhibits annual petrographic and geochemical lamination. Stable isotope analyses and heat-transfer modelling indicate that its $\delta^{18}$O oscillations are dominated by surface temperature changes transmitted to the subsurface via heat conduction (Koltai et al., 2017). Although $\delta^{18}$O provides a proxy for seasonal fluctuations in surface temperature (Koltai et al., 2017), its variability on a multi-annual time scale is well replicated by LAS 19, implying that the two flowstones were deposited close to isotopic equilibrium (Dorale and Liu, 2009).

As the well-developed petrographic lamination is due to traces of varying amounts of humic and fulvic acids, the lack of such regular laminae can be used as an indirect proxy for the depth of a given fracture. Thus we assume that non-laminated flowstones formed at greater depths and the temperature in these subsurface fractures most likely reflects the outside mean annual air temperature. Spötl et al. (2002) reported that none of the nearby perennial springs showed intra-annual temperature variability, supporting this assumption. The water temperature of such a spring at the SQ site was constant during (+12.8 ± 0.1$^{o}$C) the two-year-long monitoring period. Water dripping from the slope breccia at KO, however, showed a 3.7$^{o}$C variability (Spötl et al., 2002) which can be explained by the seasonal influence of the outside air. As we assume that the fractures were not influenced by seasonal changes in air temperature the $\delta^{18}$O signal of the Vinschgau flowstones is primarily regarded as a proxy for $\delta^{18}$O of groundwater and local precipitation. In mid and high latitudes a well-established relationship exists between air temperature and the oxygen isotopic composition of precipitation, i.e. a temperature rise of 1$^{o}$C leads to 0.59±0.09‰ higher isotope values (Rozanski et al., 1992). This would be partially counterbalanced in the speleothem $\delta^{18}$O signal by the isotope fractionation during calcite/aragonite formation. The temperature dependence of the oxygen isotope fractionation during calcite precipitation is -0.24 ‰/$^{o}$C based on experimental studies (Kim and O`Neil, 1997), while a somewhat higher value (-0.18 ‰/$^{o}$C) was determined by a cave-based study (Tremaine et al., 2011). Kim et al. (2007) reported a similar value (-0.22 ‰/$^{o}$C) for the temperature coefficient for the oxygen isotope fractionation of aragonite. Consequently, for the study area a positive (negative) net isotope change is expected in the speleothem $\delta^{18}$O signal during climate amelioration (deterioration). This signal, however, may be modified to a variable extent by in-aquifer processes as discussed in 5.2.

## 5.1.2 $\delta^{13}$C

The interpretation of the carbon isotope signal of speleothems from karst caves is commonly more challenging than that of $\delta^{18}$O (e.g. McDermott, 2004), since $\delta^{13}$C values can be affected by a variety of processes including carbon dynamics of the soil and epikarst, subsurface air ventilation, and associated disequilibrium isotope fractionation. Today, the Sonnenberg slope is mainly covered by sandy pararendzinas (Florineth, 1974) and contains a semi-arid vegetation. The strong soil moisture deficit on the slopes (Della Chiesa et al., 2014) may limit the amount of solutes entering the fracture system (Fairchild and Baker, 2012). Additionally, some of the carbonate is derived from the crystalline host rock, in particular local occurrences of Fe-carbonates (Spötl et al., 2002).

Although carbon isotopes mostly follow the fluctuations of $\delta^{18}$O, this relationship can vary within a given sample (e.g. LAS 19). LAS 19 shows a $\delta^{18}$O rise of 3.2 ‰ at the YD-Holocene transition, followed by a similar decrease in carbon isotopes (Suppl. Fig. 4), suggesting that during certain time intervals carbon isotopes may

reflect a soil signal, whereby lower $\delta^{13}C$ values correspond to an increase in soil bioproductivity (Genty et al.,
2001; Fairchild and Baker 2012; Borsato et al, 2015). As discussed below this signal is, however, masked by in-
aquifer processes.
**5.2 Aquifer-internal processes**
In a subsurface fracture system like the Sonnenberg where speleothems form as vein-filling calcite and
aragonite, prior calcite precipitation (PCP) and/or prior aragonite precipitation (PAP) are expected to influence
the carbon isotope composition of speleothems (e.g. Fairchild and Baker, 2012). Yet, recent laboratory
experiments indicate that PCP has an effect on the oxygen isotope systematics of the precipitating calcite as well
(Polag et al., 2010; Dreybrodt and Scholz, 2011). Thus we propose that PCP may lead to progressively higher
oxygen isotope values along the flow path, even if this change is of much smaller amplitude than that of $\delta^{13}C$.
Although similar experiments investigating the influence of PAP on the stable isotope composition of the
precipitating aragonite are lacking, a simultaneous enrichment in $^{13}C$ and $^{18}O$ may be expected. Evaporation-
induced disequilibrium fractionation is also likely to have an effect on the stable isotope values of calcite and
aragonite flowstones since evaporation and associated $CO_2$ degassing are the primary drivers of secondary
carbonate deposition. Evaporation exerts a significant control on oxygen isotope values, resulting in the
enrichment of $^{18}O$ in the remaining water and consequently higher $\delta^{18}O$ levels in the calcite/aragonite, while $CO_2$
degassing affects carbon levels. Stable isotope and temperature monitoring of a shallow underground pool and
its associated actively forming calcite speleothem indicates that calcite precipitation occurs close to isotopic
equilibrium with respect to $\delta^{18}O$, while $\delta^{13}C$ levels strongly deviate from equilibrium (Spötl et al., 2002). This is
also supported by the fact that even though carbon and oxygen isotopes show a covariance in several flowstones,
calcite samples and LAS 1 do not exhibit co-varying $\delta^{13}C$ and $\delta^{18}O$ values. Similarities in the absolute $\delta^{18}O$
values of the three TR samples (LAS 1, LAS 2 and LAS 21) further corroborate this, suggesting that despite
PCP/PAP occurred along the flow path, flowstone precipitation occurred close to isotopic equilibrium.
Therefore, we propose that disequilibrium fractionation likely had a negligible influence on the $\delta^{18}O$ of
speleothem calcite.
In contrast, given that $\delta^{13}C$ and $\delta^{18}O$ values co-vary in the aragonite samples (LAS 34 and LAS 72) and also in
LAS 2 independent of its mineralogy (Fig. 3), these samples may have formed out of isotopic equilibrium.
Although the covariance of carbon and oxygen isotope values may result from in-aquifer processes, it has been
widely used as an indicator of disequilibrium isotope fractionation (Hendy, 1971). The slope of regression of
$\Delta\delta^{13}C/\Delta\delta^{18}O$ varies between 2.5 and 3.4 in LAS 2, 24 and 72 (Fig. 3A). Such values indicate that disequilibrium
isotope fractionation occurred during aragonite precipitation, whereby $CO_2$ hydration and hydroxylation
reactions promoting oxygen isotope exchange between $HCO^{3}$- reservoir and $H_2O$ were not fast enough to
maintain isotopic equilibrium (Mickler et al., 2006). The lack of a similar strong correlation between the two
isotopes in the flowstone samples dominated by calcite (Fig. 3B) except for LAS 2 further supports the influence
of disequilibrium isotope effects rather than of in-aquifer processes. As none of these springs is presently
precipitating aragonite, it is hard to distinguish whether evaporation and PCP/PAP or disequilibrium isotope
fractionation or a combination of these processes has led to the covariation between $\delta^{18}O$ and $\delta^{13}C$ in the
flowstones.
To further analyse the potential influence of disequilibrium processes and local hydrology on the isotopic
composition of the Vinschgau flowstones, coeval sections were compared. Unfortunately, in all cases the

common time window of deposition is too short to provide reliable analyses using statistical methods (e.g. Fohlmeister, 2012). Nevertheless, as the deposition of the three TR samples partly overlaps, intra-fracture variability can be tested on decadal to centennial scales despite differences in growth rate and hence proxy data resolution. Given the similarities in the range of $\delta^{18}O$ variability (Fig. 4), mineralogy, and the lack of fluorescent lamination, we suggest that flowstones in a given fracture form under very similar conditions and therefore most probably record the local climate signal. Differences in the absolute values of $\delta^{18}O$ and $\delta^{13}C$ are attributed to PAP and PCP.

A stronger influence of the local hydrology and hence a lower climate signal/noise ratio is expected when comparing flowstones from different sites (Fig. 5). Due to the inter-fracture variance in PCP/PAP calcite was deposited at KO (sample LAS 19), while at the same time aragonite formed at the SQ site (sample LAS 72). Still, their $\delta^{18}O$ pattern shares some similarities within the combined errors of the two age models, while $\delta^{13}C$ in the aragonite specimen shows a much larger amplitude (8.9‰) than in the coeval calcitic one (3.6‰). This most likely reflects the combined influence of disequilibrium isotopic fractionation and the difference in the fractionation factor between the two polymorphs (Morse and Mackenzie, 1990; Frisia et al., 2002). Frisia et al. (2002) reported that carbon isotopes are 2 to 3.4 ‰ higher in aragonite than in calcite at Grotte de Clamouse (S France). $\delta^{13}C$ values similar to that of LAS 72 were reported from modern aragonite from Obstanser Eishöhle (2.4 to 7.0 ‰), an alpine cave in southern Austria (Spötl et al., 2016). Moreover, PCP (PAP) may have further increased the $\delta^{13}C$ values in the Vinschgau sites.

**5.3 Potential as a palaeoclimate archive**

Similar to speleothems from karst caves in semiarid settings (e.g. Avigour et al., 1992; McMillan et al., 2005; Hoffmann et al., 2016), fracture-filling flowstone from non-carbonate, climate-sensitive settings may provide a useful record of palaeoaridity and palaeohydrology. Annually laminated flowstones (e.g. LAS 6) that formed in a few meters depth may provide insights into changes in seasonality (Koltai et al., 2017), while those from deeper fractures likely record changes on multi-decadal to centennial resolution and thus provide a fragmented archive of local climate history. This case study shows, however, that fracture-filling speleothems also record the inherent heterogeneity of such fractured aquifers which may mask short-term climate signals.

As aragonite is metastable at Earth`s surface conditions and hence susceptible to diagenetic transformation, the possible alteration of the geochemical signal has to be considered (e.g. Domínguez-Villar et al., 2017 and references therein). Moreover, Lachniet (2015) emphasised that the $\delta^{18}O$ variability of aragonite speleothems should only be used as a proxy if aragonite precipitation occurred close to isotopic equilibrium. Thin section and XRD analyses indicate pristine aragonite in LAS 1, 2, 34 and 72. Aragonite preservation in these samples is further supported by the fact that all $^{230}Th$ ages are in stratigraphic order regardless of mineralogy (Suppl. Table 1). Nevertheless, flowstone deposition was most probably influenced by disequilibrium isotope fractionation as suggested by the high correlation between carbon and oxygen isotopes in LAS 2, 34 and 72 (Fig. 3). Therefore, $\delta^{18}O$ variability should be interpreted carefully in these three samples.

Moreover, the timing of aragonite deposition does not show any systematic relationship between the samples during the Lateglacial and the Early Holocene (Figs. 4-6). Instead petrographic analyses and hydrochemistry data of modern springs (Spötl et al., 2002) suggest that due to the high degree of total dissolved solids only small changes in water chemistry give rise to either aragonite or calcite precipitation, partly reflecting the heterogeneity of the fractured aquifer. Similarly, changes in growth rate are first and foremost driven by in-

aquifer processes including PCP and/or PAP, as indicated by the TR samples (Fig. 4). Therefore, calcite-
aragonite transitions and growth rate changes do not necessarily reflect an external (climate) signal, unless
coeval samples show a coherent pattern.
Carbon isotope data suggest a weak soil-derived signal for short time periods only (e.g. at the YD-to-Holocene
transition in LAS 19, Suppl. Fig. 4), while most values suggest buffering by inorganic carbon in conjunction
with kinetic isotope enrichment (Spötl et al., 2002).
The most prominent feature of $\delta^{18}O$ proxy record is the ~3.2‰ rise in LAS 19 at the YD-Holocene transition
(Fig. 6e). Moreover, the first order pattern of the two aragonite samples covering the Bølling-Allerød warm
phase shows a close resemblance to the $\delta^{18}O$ variability of the ostracod record from Mondsee (Lauterbach et al.,
2011), a lake in central Austria, suggesting that centennial- to orbital-scale large-amplitude changes of the
Northern Hemisphere climate system are recorded in the $\delta^{18}O$ variability of this archive.
During the YD a gradual ~1.7‰ decline in $\delta^{18}O$ is observed in LAS 21 between 12.2 and 11.7 ka BP. Parts of
this shift are also captured by LAS 1 and LAS 19, implying a related cause. Several terrestrial archives across
Europe record a change in regional climate mid-way through the YD (e.g. Brauer et al., 2008; Bakke et al., 2009;
Baldini et al., 2015; Belli et al., 2017). The onset of this transition was time-transgressive (~12.45 to 12.15 ka
BP) across Europe due to the gradual northward shift of the polar front driven by the resumption of the North
Atlantic overturning (Lane et al., 2013; Bartolomé et al., 2015). Whether this shift towards lower values in our
$\delta^{18}O$ record corresponds to the change in the regional climate or to in-aquifer processes remains unclear given
the lack of a flowstone sample covering the entire YD.
**6 Conclusions**
Petrographic and geochemical analyses of vein-filling calcite and aragonite flowstones in near-surface fractures
indicate that the latter polymorph is more susceptible to disequilibrium processes regarding both $\delta^{18}O$ and $\delta^{13}C$.
The two most important in-aquifer processes modifying the geochemical signature of these speleothems are
evaporation and PCP/PAP. Both of these processes are likely to govern variations in speleothem mineralogy, as
indicated by the deposition of coeval aragonite and calcite flowstones. Accordingly, changes in speleothem
mineralogy cannot be used to constrain the timing of past episodes of high vs. low precipitation in the
Vinschgau.
$\delta^{18}O$ variability has proved to be the most reliable climate proxy in the Vinschgau flowstones. Low-amplitude,
high-frequency (decadal-scale) variability in LAS 1, 6, 10, 19 and 21 is attributed to in-aquifer processes, while
the centennial-scale variability shows significant variation (e.g. 3.2‰) suggesting changes in the $\delta^{18}O$ of
precipitation. Although local factors, such as strong evaporation and PCP/PAP can amplify these climate
signatures, the $\delta^{18}O$ values show a broadly similar pattern to regional $\delta^{18}O$ lacustrine records (e.g. Mondsee).
Due to the lack of long overlapping sections of speleothem growth and the complexity of in-aquifer processes
this case study shows that it is highly challenging to establish a robust stacked $\delta^{18}O$ record of local climate
change on multi-millennial to orbital timescales using such speleothems. However, it is possible that fracture-
filling calcite and/or aragonite from other areas may have a high potential as a climate archive if the local
hydrogeological conditions are well constrained. Our study also emphasises that a tight age control and a multi-
proxy approach are essential in the study of such non-karstic settings.

**Acknowledgements**

This project was supported by the Autonome Provinz Bozen-Südtirol (no. 16/40.3). D. Schmidmair is acknowledged for XRD analyses and K. Wendt for linguistic help and valuable comments. We thank Ian J. Fairchild, Dana C. Riechelmann and an anonymous reviewer for constructive and thorough comments that greatly helped to improve the manuscript.

**Appendix and Supplementary data**

Stable isotope data reported in this article can be found on the NOAA website.

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

**Table 1. Stable isotope composition of the Vinschgau flowstones.**

| Sample | Mineralogy | δ$^{18}$O (‰) | | | δ$^{13}$C (‰) | | |
|---|---|---|---|---|---|---|---|
| | | min. | max. | Mean | min. | max. | Mean |
| *TR* | | | | | | | |
| LAS 1 | calcite-aragonite | -12.8 | -11.1 | -11.9 | -2.2 | 4.3 | -0.2 |
| LAS 2 | calcite-aragonite | -13.0 | -9.7 | -11.7 | -2.3 | 7.3 | 0.9 |
| LAS 21 | calcite-aragonite | -13.1 | -9.8 | -12.1 | -2.3 | 2.4 | -0.7 |
| *SQ* | | | | | | | |
| LAS 6 | calcite | -14.1 | -11.8 | -13.2 | -2.9 | -0.3 | -2.1 |
| LAS 34 | aragonite | -12.8 | -9.9 | -11.3 | -1.0 | 6.2 | 2.2 |
| LAS 72 | aragonite | -11.8 | -9.2 | -10.7 | -1.8 | 7.3 | 1.6 |
| *KO* | | | | | | | |
| LAS 10 | calcite | -12.3 | -10.3 | -11.3 | -5.6 | -1.4 | -3.2 |
| LAS 19 | calcite | -13.5 | -10.1 | -11.7 | -4.5 | -0.8 | -2.8 |

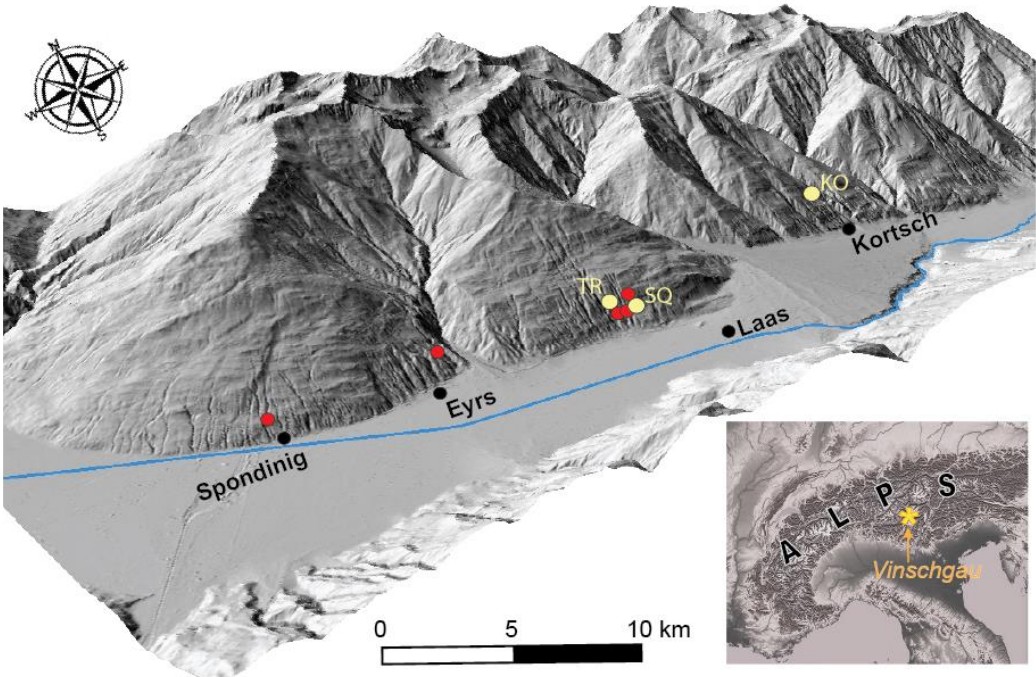

**Figure 1: Oblique view of the Vinschgau valley. Red points show the occurrence of vein-filling flowstones in the lower part of the south-facing slope (Sonnenberg). Yellow points mark the three sampling sites.**

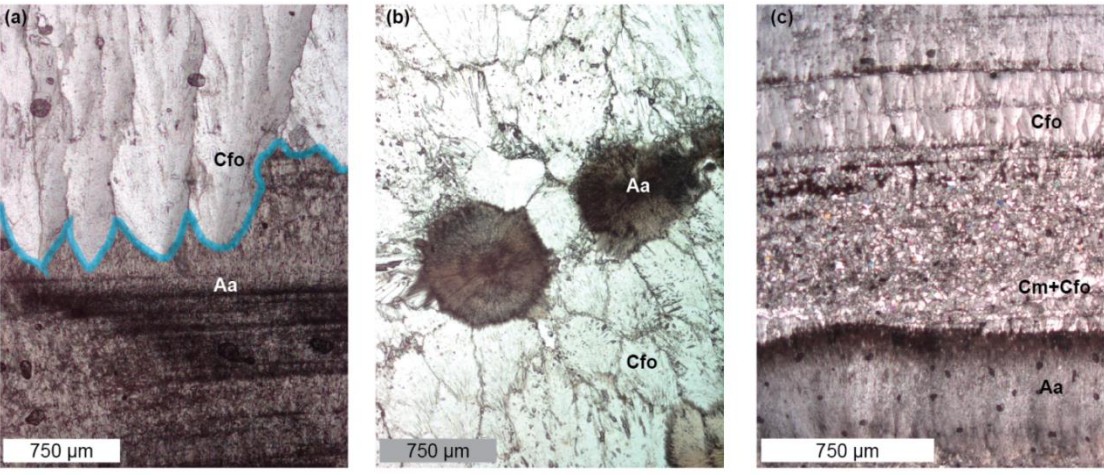

5    **Figure 2: Aragonite and calcite textures. (a) Boundary of fascicular optic calcite (Cfo) and acicular aragonite (Aa), showing competing crystal growth between the two polymorphs (blue line, sample LAS 2). (b) Co-precipitation of primary calcite and aragonite (sample LAS 21). (c) Complex fabric dominated by mosaic calcite (Cm) where fascicular optic calcite polycrystals (Cfo) are locally present between acicular aragonite (Aa) and fascicular optic calcite (Cfo, sample LAS 34).**

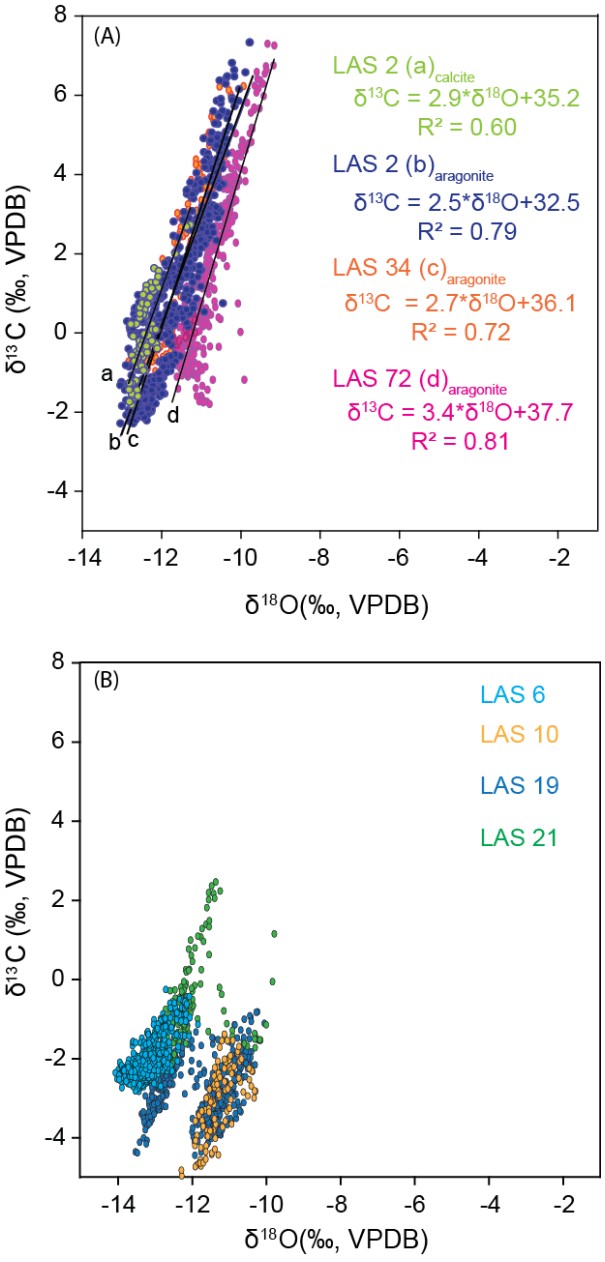

**Figure 3A: Isotope crossplot for samples LAS 2, LAS 34 and LAS 72. The highly significant correlation ($R^2 \geq 0.60$) between the two isotopes suggests strong disequilibrium-controlled isotope fractionation. B. Calcite samples only show a weak correlation between $\delta^{18}O$ and $\delta^{13}C$ ($R^2 < 0.60$).**

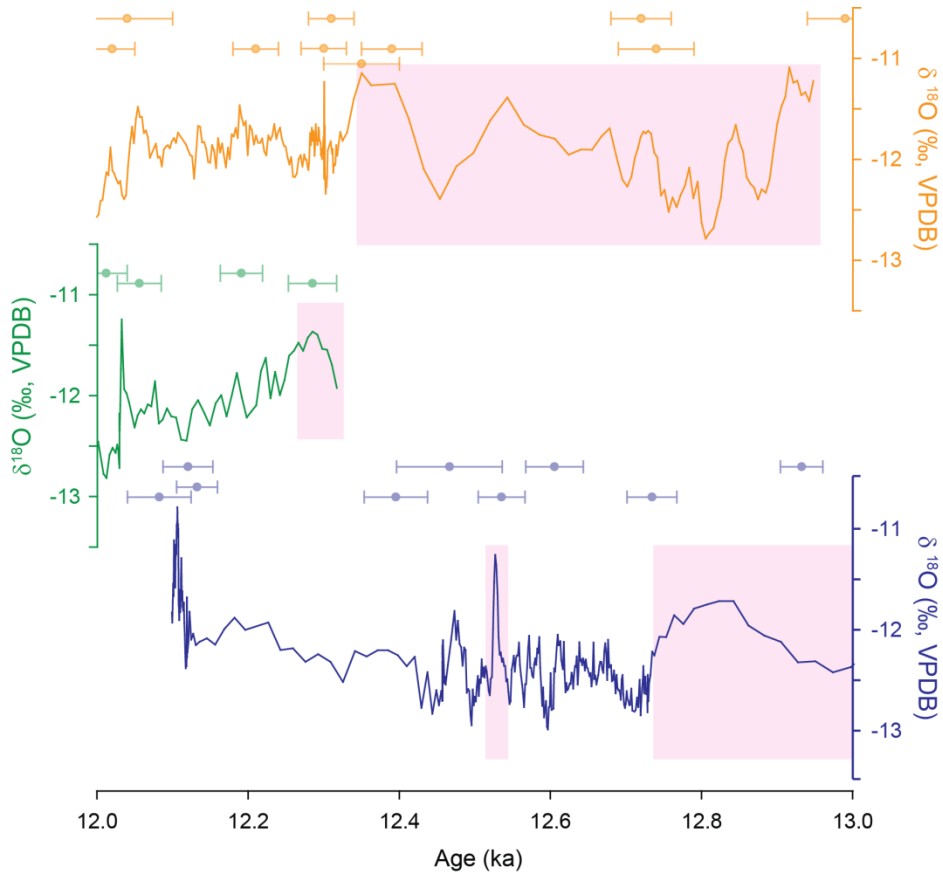

**Figure 4:** δ¹⁸O variability of the TR samples LAS 1 (orange), LAS 21 (green) and LAS 2 (blue) in their overlapping sections. All samples are plotted based on the modelled ages. [230]Th ages with their corresponding errors are plotted above each δ¹⁸O time series. Pink rectangles indicate aragonite fabric.

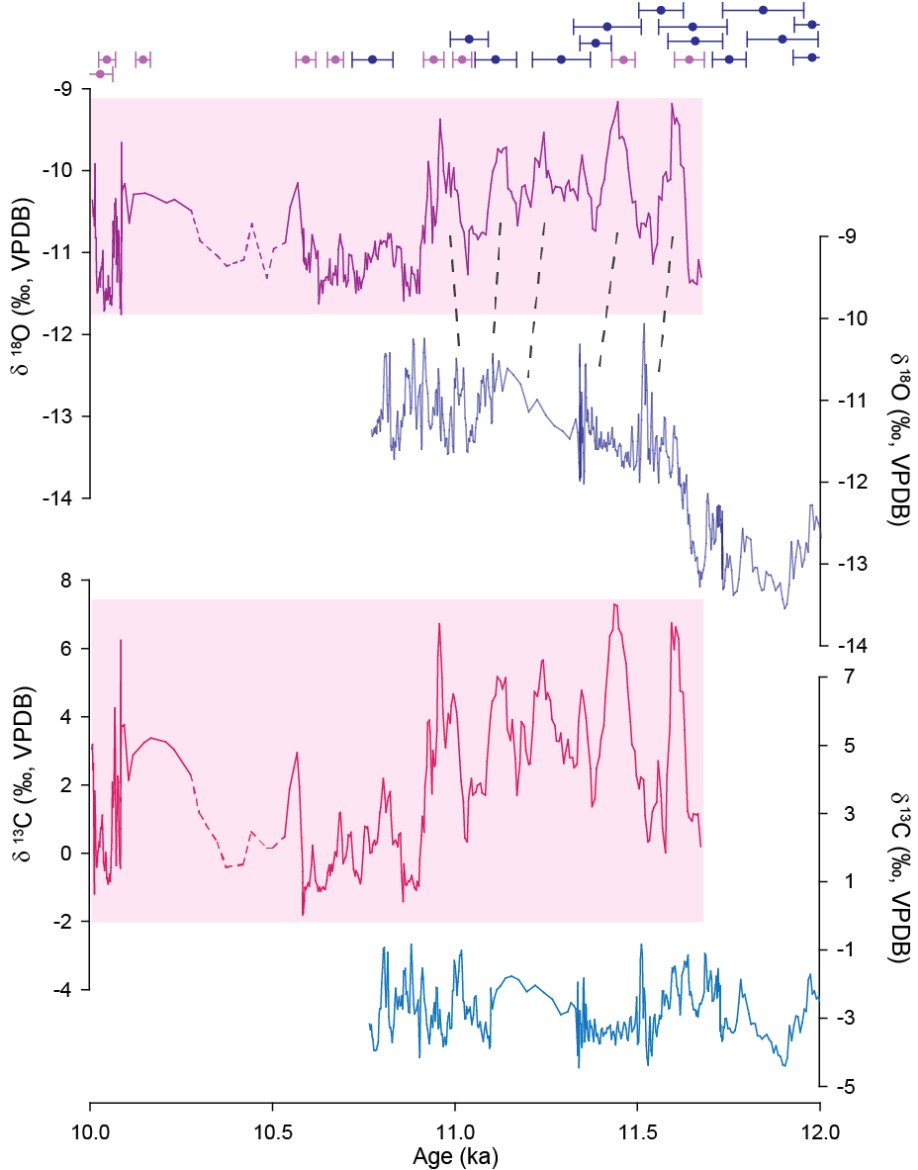

**Figure 5: Stable isotope variability of LAS 72 (pink) and LAS 19 (blue) between 12 and 10 ka, BP. Dashed lines in the stable isotope time series of LAS 72 refers to the period characterised by slow growth rate. Dashed tie lines indicate similarities between the δ$^{18}$O variability of the two flowstones. Note that LAS 72 is comprised of aragonite as indicated by the pink rectangle, while LAS 19 is a calcitic flowstone.**

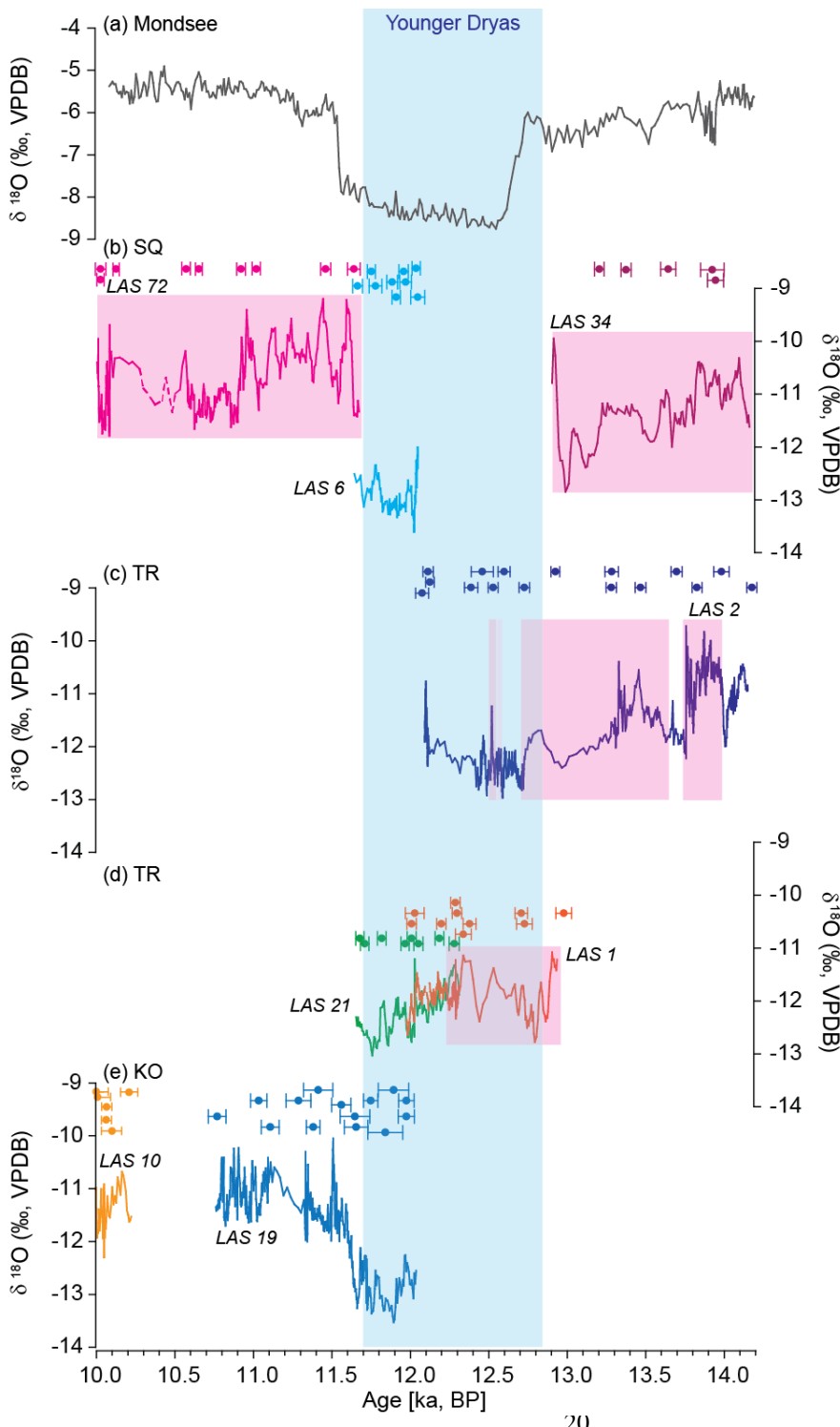

**Figure 6:** Comparison of $\delta^{18}O$ time-series of (a) benthic ostracods from Mondsee (Lauterbach et al., 2011) and the Vinschgau flowstones (b-e). (b) shows the three samples from SQ site: LAS 72 (bright pink), LAS 6 (turquoise) and LAS 34 (dark pink). The $\delta^{18}O$ variability of the TR samples is shown in (c) and (d), whereby dark blue represents LAS 2, green and orange mark the oxygen isotope record of LAS 21 and LAS 1, respectively. LAS 10 (yellow) and LAS 19 (light blue) from KO are shown in (e). Note that the pink rectangles are indicative of aragonite and the blue bar refers to the Younger Dryas.