# Peer review of "Palaeoclimate significance of speleothems in crystalline rocks"

_Climate of the Past, 2017_

## Referee Comment (RC1) · Anonymous Referee #1 · 29 Sep 2017

Authors studied 9 flowstone samples grew in non-karst host rock fractures from 3 locations in the Alps. Both aragonite and calcite were found to be present in these samples. Using U-Th geochronology, growth history was constructed and stable oxygen and carbon isotope time series were produced. Based on the covariance of oxygen and carbon isotope variations, authors believed that the oxygen isotopes are likely to be in isotopic equilibrium for all but 2 samples, and reflect local precipitation signals, while carbon isotopes are likely affected significantly by kinetic fractionation. Authors noted the similarity of segment of the record with coeval ice core oxygen isotope time series at the

beginning of Holocene and suggest that these flowstones do contain climate signals, even though a more robust and continuous record are challenging.

Overall, the study of speleothems outside a cave environment is a very attractive proposition as it greatly expands the areas for which a study can be carried out. The manuscript contains excellent work and should warrant publication. There are some edits or clarifications, however, I hope authors will consider.

First, the complete data for oxygen and carbon isotope analysis is not available in either manuscript, or supplementary documents. The data table only listed min and max isotope values for each of the nine samples. This greatly restricted exploration of the data by readers who intend to do so.

Second, the presentation of oxygen isotope record is fragmented and difficult to read. Because aragonite and calcite fractionation factors are known, it may worth converting aragonite stable isotopes values to equivalent calcite values and construct a composite record. This may produce a more readable time series, which can be more easily compared with ice core record. On these time series, it is also apparently that some samples represent much larger time windows than others (e.g., fig 5, first SQ time series, between 10-10.5 ka). For these samples, line graph masked the uncertainty in the magnitude and frequency of the variations. Authors may consider not connecting the individual data points in these locations.

Third, I have reservations on the reliability of using covariance of oxygen and carbon isotopes as an indicator of the presence of kinetic effects. As pointed out in Feng et al. (2012) and Myers et al. (2014), oxygen and carbon isotopes of calcite are affected by difference factors: PCP/PAP is the major control on calcite isotope values, where it has virtually no impact on oxygen isotopes, which is controlled by evaporation, growth rate. A strong covariance between oxygen and carbon isotopes does not necessarily indicate a significant impact of oxygen isotope kinetic fractionation.

Fourth, lack of image of the samples studied. Most speleothem samples from cave

have regular growth patterns and readers are familiar with them. A flowstone grew in the fracture of non-karst settings is more difficult to visually grasp. A photo of the sample (if it's taken as a whole) or a diagram (if a core was taken) could help readers understand the work being done.

Fifth, Large part of section 5.3 seems belong to the introduction section rather than discussion section. The introduction section, as written, is a bit light. Moving some part of the text from 5.3 to introduction may provide more background for readers before presenting the details of the study.

Lastly, author should address the problems surrounding the flowstone. Most studies avoid flowstone due to concerns of kinetic effects, what steps have authors taken to avoid this pitfall?

---

## Referee Comment (RC2) · I. J. Fairchild (Referee) · 30 Oct 2017

**Comments on Koltai et al (`cp-2017-100`) by Ian Fairchild**

**Overview**

This paper brings a little-studied type of speleothem site to the attention of the readership. Other such sites may have been overlooked up to now, but offer some interesting possibilities for palaeoclimate work particularly in areas where carbonate rocks are lacking. The likelihood of obtaining good-quality dates owing to the high U content of groundwater is a particularly attractive feature.

The authors have worked on an interesting site where speleothem deposits occur in non-carbonate fractured bedrocks, building on some previous publications at this site. The strengths of the current paper are 1) the large number of good U-Th dates demonstrating periods of overlapping growth, 2) the discussion of the relationship between mineralogy and isotope geochemistry and lack of relationship of mineralogy to growth period and 3) documentation of the extent to which the combined dataset reproduces known isotopic shifts in the Alpine region. As a result, the authors provide a balanced, cautiously optimistic view of the potential of such non-carbonate bedrock sites for future palaeoclimate work. The work is very well-presented for the most part, although a few suggestions for minor change are made below.

**Relationship with water chemistry**

Spötl et al. (2002) presented some water analyses from the study area. The water are said to be highly mineralised. Can more discussion be drawn from those data about the likely conditions of growth of the fracture-fills, e.g Ionic strength, Mg/Ca, oxidation state, supersaturation etc?

**Macroscopic nature of the precipitates**

I think it would be helpful for readers looking for analogous occurrences elsewhere to provide a figure with macroscopic images of the vein-lining deposits and to describe and comment on features at this spatial scale.

**Secondary calcite**

P4, lines 9-10 and Figure 2. I could not see why this calcite couldn't be primary from the information provided. If it has straight extinction rather than undulose extinction that might be a criterion (as would a low Mg content).

**Stable isotope compositions**

1. P6, lines 13-14. Carbonate isotopes are primarily interpreted as a proxy for isotopic composition of atmospheric precipitation, but temperature changes in the 13-10 ka intervals surely may have been sufficient to have influenced the composition of calcite too?

2. The relationship between carbon and oxygen isotopes in some samples is for a strong covariation, but the slope is not mentioned in the text. Carbon isotopes change much more than oxygen. Needs a bit more discussion, including ideas of Hansen et al (GCA, 2017) of which Spötl is a co-author. Could there be some equilibration effects here?

3. The discussion properly considers potential differences amongst the conditions in which aragonite and calcite form.
   a) However, the two phases are not distinguished in the crossplots of Figure 3 and I think it would be helpful to do this.

b) I also think that the data for the other samples should also be cross-plot to illustrate more clearly the variation in the extent of covariation.

c) LAS34 is shown in Fig. 3, but ages are not plotted in a figure – the reason for this is apparently not referred to in the text.

**Suggested minor corrections**

P1, Line 23. The words: "by yielding high-resolution, multi-proxy data" are redundant. By no means all of the iconic records are multiproxy or high-resolution.

P2, line 20 "E-W" is conventional, not "W-E"

P2, line 24 To avoid repetition of the word *valley* in "the deglaciation of the valley", how about "local deglaciation".

P2, line 29. Reference to "highly mineralised". More details would be useful.

P3, line 27. "chemical" rather than "chemistry"

P4, line 5" detritus-rich" not "detrital-rich"

P4, line 6 "however" requires a semi-colon before it and does not need the comma afterwards. Alternatively substitute "but".

P4, line 7. "thick" not "wide". [the habit of some authors of referring to the "width" of layers comes from the dendroclimatological literature, but the geometry of tree rings is different]

P4, lines 29-30. "enriched" and "depleted" Higher and lower delta values are meant. [see Sharp (2007)!]

P4, lines 33-34. The actual slope of the regression lines is not mentioned. Carbon isotopes changes much more than oxygen.

P6, line 8 "springs"

**Figure 1.** Neither this figure nor the text mention the altitude of the site or specify its precise location.

**Figure 2.** This figure (part c) does not make it clear why the secondary calcite cannot be primary since simultaneous growth of aragonite and calcite is also shown elsewhere.

**Figure 4, caption.** Superscript needed: $^{230}$Th

**Figure 5** has references to colour, but these need to be made more explicit for the colour-blind. Add a background shade for the aragonitic sample, as in Figures 4 and 6.

**Figure 6** reference to colour in the caption should be supplemented by writing the name of the sample next to the plots on the figure itself.

**Supplementary information**

Supplementary table 1. Excessive significant figures for $^{238}$U and $^{230}$Th/$^{232}$Th ratios?

It would be useful to present the STALAGE age models graphically in the supplementary information

---

## Referee Comment (RC3) · F. C. Riechelmann (Referee) · 17 Nov 2017

**Dana Riechelmann # 3:**

In this study, the authors present very well dated speleothems, which grew in the factures of a gneiss hostrock. They analysed the petrography and the $\delta^{13}C$ and $\delta^{18}O$ values of eight flowstones. This study show the potential of such carbonate deposits as palaeoclimate archives, which are however, in this case not straight to interpret in case of past climate variations, due to low time overlap between the different flowstones. However, this is a nice dataset and a very well written paper.

There are some general and minor comments and corrections to be done, which are listed below. I could recommend this manuscript, after minor revisions, for publication.

**General Comments:**

- The introduction is a bit short. The introducing part (Page 1, Line 22 to 35) could be more extended, with for example more details about the study of Frisia et al. (2017) about the analyses and results of similar studies. The part with the aim of your study (Page 1, Line 36 to Page 2 Line 17) is in relation to the introducing part quite long. Perhaps it is possible to move some of this to the methods part.

- Overview pictures of the samples would be nice for the reader. In these pictures, it would be helpful for the reader, if all datings are marked and the ages are written next to them and the parts are marked used for this study. Therefore, it is clear that you used only specific time spans and that there is more material from other times spans.

- Could you please mark the similarities of the isotope records in Figure 5. That would make it much easier for the reader to follow.

- Could the occurrence of aragonite in the flowstone be interpret as drier periods?

**Specific and technical Comments in chronological order:**

Page 1, Line 13: Please replace "kinetic" with "disequilibrium".

Page 2, Line 29: Please replace "by mass movements" with "by the mass movements".

Page 2, Line 29: Please delete the "on" at the end of the line.

Page 4, Line 8: What calcite fabric occurs in LAS 72, 1, 2, 21 and 34? Please provide this information. Are these also columnar fascicular optic?

Page 4, Line 29: Please replace "more enriched" with "higher".

Page 4, Line 30: Please replace "most depleted" with "lowest".

Page 4, Lines 33-34: I do not see two regression lines for LAS 2 in Figure 3. There should be one for the calcite part and one for the aragonite part, as I understood from the text.

Page 5, Line 6: Please replace "more negative" with "lower".

Page 5, Line 8: Please replace "more positive" with "higher".

Page 5, Lines 12-13: Please add that the referred Table 1 is the Supplement Table 1.

Page 5, Line 29: I think you mean "Suppl. Fig. 2a" instead of "Suppl. Fig. 5".

Page 6, Lines 9-10: Please give the exact temperature as a number.

Page 6, Line 14: What do you exactly mean with "…primarily regarded as a proxy for $\delta^{18}O$ of local precipitation."? There is no amount effect in the $\delta^{18}O$ of the precipitation in the Alps. Therefore, the $\delta^{18}O$ of the precipitation should have a relation to temperature. Due to the quite long transfer time of the water you mentioned, the water should contain long-term changes in the $\delta^{18}O$ of precipitation and therefore, of long-term changes in temperature. Please add some more information to this topic at this point.

Page 6, Line 19: Please replace "kinetic" with "disequilibrium".

Page 6, Line 27: Please replace "more enriched" with "higher".

Page 6, Line 29: I am not sure if a co-variation of $\delta^{18}O$ and $\delta^{13}C$ indicate in-aquifer processes. Therefore, please delete "as indicated by the co-variation of $\delta^{18}O$ and $\delta^{13}C$ isotopes.".

Page 6, Line 35: Please replace "kinetic" with "disequilibrium".

Page 7, Line 5: Please replace "kinetic" with "disequilibrium".

Page 7, Line 9: Please replace "kinetic" with "disequilibrium".

Page 7, Line 23: Please replace "kinetic" with "disequilibrium".

Page 7, Lines 34-35: Please delete "and thus provide short snapshots of local climate.". For me a multi-decadal to centennial resolution to not provide snapshots, which are some very short time-intervals for me.

Page 8, Line 9: Please replace "kinetic" with "disequilibrium".

Page 8, Line 13: You refer to Fig. 8, however, there is no Figure 8 in the manuscript.

Page 8, Lines 16-18: Please show this in a figure. That makes it easier for the reader.

Page 8, Line 20: Please refer also to the specific panel in Figure 6.

Page 8, Line 20: Please mark the Younger Dryas and the Boling-Allerod in Figure 6.

Page 8: Line 30: Please replace "more negative" with "lower".

Page 8, Line 35: Please replace "kinetic" with "disequilibrium".

Page 9, Line 2: Please specify the in-aquifer processes.

Page 9, Lines 3-4: You write here about changing hydrological condition. This could be discussed a bit more in detail in the discussion. For me it did not come up so clearly from the discussion. This is more a general comment.

Table 1: Could you perhaps mark values from aragonite and calcite in different colours? That make it straight forward for the reader.

Figure 1: Are there samples from the vein-filling flowstones from the two other sites at Sponding and Eyrs? Do they perhaps provide time-spans missing in the other and could complete the record or give a better overlap?

Figure 3: Please replace "kinetically" with "disequilibrium".

Figure 4: Please replace "230Th" with "$^{230}$Th".

Figure 6: Please replace "shown in (e) and (d)" with "shown in (d) and (e)".

Figure 6: There is something missing in "represents the of LAS2,".

Figure 6: I think the blue rectangle marks the Younger Dryas, but this is not indicated in the figure or the figure caption.

---

## Author Comment (AC1) · 28 Dec 2017

Submission of reply to the comments made by Reviewer #1
Ms. Ref. No.: CP-2017-100
Title: Palaeoclimate significance of speleothems in crystalline rocks: a test case from the Lateglacial and Early Holocene (Vinschgau, northern Italy).

**Reviewer #1**

We are grateful for the positive and helpful comments and we address below the points raised by this referee (in italics).

*First, the complete data for oxygen and carbon isotope analysis is not available in either manuscript, or supplementary documents. The data table only listed min and max isotope values for each of the nine samples. This greatly restricted exploration of the data by readers who intend to do so.*

Due to the large amount of data we prefer not to include a long separate table in the supplementary materials, unless the editor tells us otherwise. We will make the results of stable isotope analyses available on the NOAA website once the manuscript is published.

*Second, the presentation of oxygen isotope record is fragmented and difficult to read. Because aragonite and calcite fractionation factors are known, it may worth converting aragonite stable isotopes values to equivalent calcite values and construct a composite record. This may produce a more readable time series, which can be more easily compared with ice core record. On these time series, it is also apparently that some samples represent much larger time windows than others (e.g., fig 5, first SQ time series, between 10-10.5 ka). For these samples, line graph masked the uncertainty in the magnitude and frequency of the variations. Authors may consider not connecting the individual data points in these locations.*

Yes, the stable isotope record is fragmented. We decided to present the samples belonging to each fracture separately, because the figure is difficult to read if the eight time series overlap each other. We are aware of the possibility of converting the aragonite oxygen isotope values to equivalent calcite values. Our main concern about a composite $\delta^{18}O$ record is the lack of sufficiently long overlapping segments. We run the ISCAM algorithm (Fohlmeister, 2012) for all coeval samples, but due to the shortness of common depositional periods no stacked records could be built. Thus, we prefer not to construct a stack record.

The chronology of LAS 72 suggests that there may be a hiatus between $10.57 \pm 0.3$ and $10.12 \pm 0.2$ ka. In the revised manuscript separate age models are provided for the two sections of the flowstone. The text and the figures have been corrected.

*Third, I have reservations on the reliability of using covariance of oxygen and carbon isotopes as an indicator of the presence of kinetic effects. As pointed out in Feng et al. (2012) and Myers et al. (2014), oxygen and carbon isotopes of calcite are affected by difference factors: PCP/PAP is the major control on calcite isotope values, where it has virtually no impact on oxygen isotopes, which is controlled by evaporation, growth rate. A strong covariance between oxygen and carbon isotopes does not necessarily indicate a significant impact of oxygen isotope kinetic fractionation.*

We are not entirely sure if the reviewer is referring to the paper by Meyer et al. (2014) published in GCA or a different paper. Also, it is unclear to us if the reviewer means carbon isotope values instead of calcite isotope values (line 4). In our response, we consider that she/he meant carbon isotope values.

First of all, we mostly agree with the reviewer. Evaporation has a major control on oxygen isotope values, resulting in higher $\delta^{18}O$ levels in the remaining water and consequently in higher oxygen isotope values of the calcite/aragonite. We also agree that PCP/PAP has a significant influence on carbon isotopes values but has very little impact on the oxygen isotopes. Yet, as recent laboratory experiments indicate, PCP may have an effect on both isotope systems of the precipitating calcite (Polag et al., 2010; Dreybrodt and Scholz, 2011). We suggest that PCP may lead to progressively higher $\delta^{18}O$ values along the flowpath, even if this change may be much smaller in amplitude than that

of $\delta^{13}C$. Laboratory studies investigating the influence of PAP on the stable isotope composition of the precipitating aragonite are lacking, but a simultaneous enrichment in $^{13}C$ and $^{18}O$ is to be expected.

Although the covariance of carbon and oxygen isotopes values may therefore not necessarily point to disequilibrium isotope effects, it has been widely used as an indicator of kinetic isotope fractionation (e.g. Hendy, 1971). Hendy (1971) attributed this covariation to Rayleigh distillation enrichment of $^{13}C$ and $^{18}O$ in the $HCO_3^-$ reservoir due to $CO_2$ degassing and secondary carbonate precipitation. Even though the "Hendy test" has been criticised (e.g. Dorale and Liu, 2009), we propose that in samples LAS 2, LAS 34 and LAS 72 the observed correlation between the two isotopes most probably suggests strong disequilibrium isotope fractionation based on the arguments below.

(1) The slope of regression of $\Delta\delta^{13}C/\Delta\delta^{18}O$ varies between 2.7 and 3.4. This indicates that disequilibrium isotope fractionation occurred during aragonite precipitation, whereby $CO_2$ hydration and hydroxylation reactions promoting oxygen isotope exchange between $HCO_3^-$ reservoir and $H_2O$ were not fast enough to maintain isotopic equilibrium (cf. Mickler et al. 2006). The theoretical model of Guo et al (2009) calculates a slope of 2.3 when $CaCO_3$ precipitation, dehydration and $CO_2$ degassing were dominant.

(2) Coeval flowstones dominated by calcite do not show such a strong correlation between $\delta^{13}C$ and $\delta^{18}O$.

(3) Monitoring of springs in the Vinschgau suggests that calcite precipitation occurs close to isotope equilibrium with respect to $\delta^{18}O$ (Spötl et al., 2002), but note that at none of these springs aragonite is forming today. On the other hand, $\delta^{13}C$ values of both modern speleothems and DIC show difference between pool and vadose settings, pointing towards more pronounced $CO_2$ degassing and related disequilibrium isotope fractionation in the vadose zone.

Thus, without coeval flowstones showing a consistent signal, the $\delta^{18}O$ variability of the aragonite samples should be treated with caution. In the revised manuscript we provide further discussion on this topic.

*Fourth, lack of image of the samples studied. Most speleothem samples from cave have regular growth patterns and readers are familiar with them. A flowstone grew in the fracture of non-karst settings is more difficult to visually grasp. A photo of the sample (if it's taken as a whole) or a diagram (if a core was taken) could help readers understand the work being done.*

Good point. An additional figure (Suppl. Fig. 1) showing the hand specimens has been prepared.

*Fifth, Large part of section 5.3 seems belong to the introduction section rather than discussion section. The introduction section, as written, is a bit light. Moving some part of the text from 5.3 to introduction may provide more background for readers before presenting the details of the study.*

In response to the reviewer`s suggestion both the Introduction and section 5.3 of the Discussion have been modified.

*Lastly, author should address the problems surrounding the flowstone. Most studies avoid flowstone due to concerns of kinetic effects, what steps have authors taken to avoid this pitfall?*

In the Vinschgau, speleothems form exclusively as flowstones. We evaluated our data critically and assessed if calcite and aragonite deposition occurred in isotopic equilibrium (5.1. Stable isotope systematics). Secondly, the influence of PCP/PAP and evaporation on the different speleothem proxies (stable isotopes, growth rate, mineralogy) is also discussed in detail (under 5.2. Aquifer-internal processes). Thirdly, we compared changes in mineralogy and oxygen isotope composition between coeval flowstones (both from the same fracture and from different fractures) to provide replication tests as suggested by Dorale and Liu (2009). Given that the $\delta^{18}O$ variability of the coeval carbonates is comparable (e.g. LAS 6 and LAS 19) we argue that these samples reflect primarily an external (climate) signal.

Lastly, we would like to point out that similarly to stalagmites flowstones have been shown to be valuable palaeoclimate archives (e.g. Baker et al., 1995; Holzkämper et al., 2005; Drysdale et al., 2006; Boch and Spötl, 2011; Meyer et al., 2012; Regattieri et al., 2014; Koltai et al., 2017).

Yours sincerely,

Gabriella Koltai
(on behalf of all co-authors)

---

## Author Comment (AC2) · 28 Dec 2017

Submission of reply to the comments made by Reviewer #2
Ms. Ref. No.: CP-2017-100
Title: Palaeoclimate significance of speleothems in crystalline rocks: a test case from the Lateglacial and Early Holocene (Vinschgau, northern Italy).

(reviewer text in italics)

**Reviewer #2**

We appreciate the positive and constructive comments which helped to improve this manuscript. We went through the detailed comments (e.g. typos, wording) and changed the manuscript accordingly.

*Relationship with water chemistry*
*Spötl et al. (2002) presented some water analyses from the study area. The water are said to be highly mineralised. Can more discussion be drawn from those data about the likely conditions of growth of the fracture-fills, e.g Ionic strength, Mg/Ca, oxidation state, supersaturation etc?*

In the Vinschgau, modern springs relevant for this study are characterised by slow discharge (< 1 l/sec), high electric conductivity (720-2300 µS/cm) and elevated Mg/Ca molar ratios varying from 0.25 to 3.28. All springs are supersaturated with respect to calcite and also to aragonite except for one. SI values reach as high as +1.2 and +1.0 for calcite and aragonite, respectively.

Section 2.1 has been extended to provide more information on water chemistry, and for further data we refer to the original research paper by Spötl et al., 2002.

*Macroscopic nature of the precipitates*
*I think it would be helpful for readers looking for analogous occurrences elsewhere to provide a figure with macroscopic images of the vein-lining deposits and to describe and comment on features at this spatial scale.*

A new figure (Suppl. Fig. 1) showing the hand specimens of the eight vein-lining deposits has been prepared.

*Secondary calcite*
*P4, lines 9-10 and Figure 2. I could not see why this calcite couldn't be primary from the information provided. If it has straight extinction rather than undulose extinction that might be a criterion (as would a low Mg content)*

We re-investigated the discussed thin section (LAS 2) and found indication of competitive growth of aragonite and calcite. Both the text and Fig. 2. have been corrected accordingly.

*Stable isotope compositions*
*1. P6, lines 13-14. Carbonate isotopes are primarily interpreted as a proxy for isotopic composition of atmospheric precipitation, but temperature changes in the 13-10 ka intervals surely may have been sufficient to have influenced the composition of calcite too?*

Thank you for pointing it out that this paragraph may have been misleading. Yes, changes in ambient temperature and therefore in fracture temperature most probably have had an influence on the speleothem isotopes. Speleothem $\delta^{18}O$ values are a function of cave air temperature and the $\delta^{18}O$ value of the carbonate-precipitating drip water (e.g. Lachniet, 2009).

We interpret the oxygen isotopes in our record primarily as a proxy of $\delta^{18}O$ in precipitation. A temperature rise of $1°C$ would lead to 0.59±0.09‰ higher isotope values in precipitation in the mid and high latitudes (Rozanski et al., 1992). This would be partially counterbalanced by the isotope

fractionation when calcite/aragonite forms. The temperature dependence of the oxygen isotope fractionation during calcite precipitation is -0.24 ‰/$^o$C based on experimental studies (Kim and O`Neil, 1997), while a somewhat higher value (-0.18 ‰/$^o$C) was determined by a cave-based study (Tremaine et al., 2011). Kim et al. (2007) reported a similar value (-0.22 ‰/$^o$C) for the temperature coefficient for the oxygen isotope fractionation of aragonite. Consequently, for the study area a net isotope change of 0.35-0.41‰/$^o$C and 0.37‰/$^o$C is expected for calcite and aragonite, respectively.

In response to the reviewer`s comment we rewrote this part of the discussion to avoid confusion.

*2. The relationship between carbon and oxygen isotopes in some samples is for a strong covariation, but the slope is not mentioned in the text. Carbon isotopes change much more than oxygen. Needs a bit more discussion, including ideas of Hansen et al (GCA, 2017) of which Spötl is a co-author. Could there be some equilibration effects here?*

In response to the reviewer`s suggestion we improved the paragraph by discussing the slope of the regression lines. Although studies for aragonite are lacking, laboratory experiments on calcite precipitates resulted in a regression slope of $\Delta\delta^{13}C/\Delta\delta^{18}O$ of 1.4 ± 0.6 (Wiedner et al., 2008) in case of fast degassing, while a modelling suggested that during slow degassing the slope of $\Delta\delta^{13}C/\Delta\delta^{18}O$ should approach an infinite value (Mickler et al., 2006). The authors do not see a reason to include the study by Hansen et al. (2017) in the discussion which only refers to calcite. The evidence of non-equilibrium isotope fractionation is compelling.

The aragonite phases of the LAS samples exhibit slopes of regression between 2.7 and 3.4. This indicates that disequilibrium isotope fractionation occurred during aragonite precipitation, whereby $CO_2$ hydration and hydroxylation reactions promoting oxygen isotope exchange between $HCO_3^-$ reservoir and $H_2O$ were not fast enough to maintain isotopic equilibrium (cf. Mickler et al. 2006).

*3. The discussion properly considers potential differences amongst the conditions in which aragonite and calcite form.*
*a) However, the two phases are not distinguished in the crossplots of Figure 3 and I think it would be helpful to do this.*
*b) I also think that the data for the other samples should also be cross-plot to illustrate more clearly the variation in the extent of covariation.*

LAS 34 and LAS 72 are aragonite samples. Aragonite and calcite are only present in LAS 2. Initially we did not distinguish the two polymorphs graphically, because the two slopes are very similar to each other: $\delta^{13}C = 2.85*\delta^{18}O + 35.2$ (calcite) and $\delta^{13}C = 2.70*\delta^{18}O + 32.4$ (aragonite). $R^2 = 0.60$ and 0.79 for calcite and aragonite, respectively. Following the reviewer`s suggestion Fig. 3 was modified and isotopic cross plots for other samples were included.

*c) LAS34 is shown in Fig. 3, but ages are not plotted in a figure – the reason for this is apparently not referred to in the text.*
We do not fully understand what the reviewer is referring to. The ages of LAS 34 are plotted both in Fig. 5 and Suppl. Fig. 2 and the growth period of LAS 34 is also mentioned in the text (p. 5. lines 24-25).

***Suggested minor corrections***

*P1, Line 23. The words: "by yielding high-resolution, multi-proxy data" are redundant. By no means all of the iconic records are multiproxy or high-resolution.*
Corrected

*P2, line 20 "E-W" is conventional, not "W-E"*
Corrected

*P2, line 24 To avoid repetition of the word valley in "the deglaciation of the valley", how about "local deglaciation".*
Corrected

*P2, line 29. Reference to "highly mineralised". More details would be useful.*
Section 2.1 was expanded.

*P3, line 27. "chemical" rather than "chemistry"*
Corrected

*P4, line 5" detritus-rich" not "detrital-rich"*
Corrected

*P4, line 6 "however" requires a semi-colon before it and does not need the comma afterwards. Alternatively substitute "but".*
Corrected

*P4, line 7. "thick" not "wide". [the habit of some authors of referring to the "width" of layers comes from the dendroclimatological literature, but the geometry of tree rings is different]*
Corrected

*P4, lines 29-30. "enriched" and "depleted" Higher and lower delta values are meant. [see* Sharp (2007)!]
Corrected

*P4, lines 33-34. The actual slope of the regression lines is not mentioned. Carbon isotopes changes much more than oxygen.*
This paragraph has been improved (please see discussion above).

*P6, line 8 "springs"*
Corrected

**Figure 1.** *Neither this figure nor the text mention the altitude of the site or specify its precise location.*
The precise locations of the fractures have been added to the text.

**Figure 2.** *This figure (part c) does not make it clear why the secondary calcite cannot be primary since simultaneous growth of aragonite and calcite is also shown elsewhere.*
As mentioned above, we re-examined the thin section showed in Fig. 2c and excluded the presence of secondary calcite. Both the figure and the text have been corrected accordingly.

**Figure 4, caption.** *Superscript needed:* $^{230Th}$.
Corrected

**Figure 5** *has references to colour, but these need to be made more explicit for the colour-blind. Add a background shade for the aragonitic sample, as in Figures 4 and 6.*
Thank you for the suggestion. Pink background colour has been added.

**Figure 6** *reference to colour in the caption should be supplemented by writing the name of the sample next to the plots on the figure itself.*
Sample names have been added to the figure and the figure caption has been modified.

***Supplementary information***
*Supplementary table 1. Excessive significant figures for 238U and 230Th/232Th ratios?*
*It would be useful to present the STALAGE age models graphically in the supplementary information*

StalAge models were graphically already presented in the original version of the manuscript as Suppl. Figs. 1-2.

Yours sincerely,

Gabriella Koltai
(on behalf of all co-authors)

---

## Author Comment (AC3) · 28 Dec 2017

Submission of reply to the comments made by Reviewer #3
Ms. Ref. No.: CP-2017-100
Title: Palaeoclimate significance of speleothems in crystalline rocks: a test case from the Lateglacial and Early Holocene (Vinschgau, northern Italy).

**Reviewer #3**

We thank the reviewer for providing thorough and constructive comments on our manuscript. Below we address each comment individually.

(reviewer text in italics)

*General Comments:*
*The introduction is a bit short. The introducing part (Page 1, Line 22 to 35) could be more extended, with for example more details about the study of Frisia et al. (2017) about the analyses and results of similar studies. The part with the aim of your study (Page 1, Line 36 to Page 2 Line 17) is in relation to the introducing part quite long. Perhaps it is possible to move some of this to the methods part.*

The introduction has been extended, but we have not shortened the description of the aims. We feel that the current description is appropriate in order to explain why these eight vein-lining deposits (out of a total of ca. 70 samples) were chosen for this study.

*Overview pictures of the samples would be nice for the reader. In these pictures, it would be helpful for the reader, if all datings are marked and the ages are written next to them and the parts are marked used for this study. Therefore, it is clear that you used only specific time spans and that there is more material from other times spans.*

A new figure (Suppl. Fig.1) showing the analysed hand specimens has been prepared. It is also indicated in the figure which section of each speleothem was examined in this study. We decided not to add the $^{230}$Th ages because this would overload the images.

*Could you please mark the similarities of the isotope records in Figure 5. That would make it much easier for the reader to follow.*

Fig. 5 has been modified for a better understanding. Dashed lines were added to show the similarities between the oxygen isotope records and the aragonite fabric is indicated by a pink background colour.

*Could the occurrence of aragonite in the flowstone be interpret as drier periods?*

This is a very good point but we think that the reviewer might have missed this part in the text.

In our study aragonite should not be interpreted as a palaeoaridity indicator. A considerable effort was put into U-Th dating and petrographic analyses of these samples. The results do not show any systematic relationship between speleothem mineralogy and climate during the Lateglacial and the Early Holocene. Fig. 4 illustrates an example showing that while sample LAS 1 was deposited as an aragonite flowstone from ca. 13.0 to 12.3 ka, the coeval LAS 2 sample mostly formed as calcite.

Instead, petrographic analyses and hydrochemistry data of modern springs (Spötl et al., 2002) suggest that due to the high degree of total dissolved solids only a small change in water chemistry gives rise to either aragonite or calcite precipitation, partly reflecting the heterogeneity of this fractured aquifer. Accordingly, changes in speleothem mineralogy cannot be used to constrain the timing of past episodes of high vs. low precipitation.

In Section 2.3 we state (p.8 lines 12-15): "changes in speleothem mineralogy and growth rate are first and foremost driven by in-aquifer processes including PCP and/or PAP, as indicated by the TR samples and LAS 19 and LAS 72 (Figs. 4 and 6). Therefore, calcite-aragonite transitions and growth rate changes do not necessarily reflect an external (climate) signal, unless coeval samples show a coherent pattern."

A sentence has been added to the Conclusions to emphasise this result.

***Specific and technical Comments in chronological order:***

*Page 1, Line 13: Please replace "kinetic" with "disequilibrium".*
Corrected

*Page 2, Line 29: Please replace "by mass movements" with "by the mass movements".*
An article has been added.

*Page 2, Line 29: Please delete the "on" at the end of the line.*
We think deleting "on" would slightly change the meaning of this sentence, therefore we prefer not to change this.

*Page 4, Line 8: What calcite fabric occurs in LAS 72, 1, 2, 21 and 34? Please provide this information. Are these also columnar fascicular optic?*
Yes, the characteristic calcite fabric in these samples is also columnar fascicular optic calcite. This information was added in the text.

*Page 4, Line 29: Please replace "more enriched" with "higher".*
Corrected

*Page 4, Line 30: Please replace "most depleted" with "lowest".*
Corrected

*Page 4, Lines 33-34: I do not see two regression lines for LAS 2 in Figure 3. There should be one for the calcite part and one for the aragonite part, as I understood from the text.*
Yes, there is only one regression line for LAS 2 in Fig. 3. We did not distinguish the two polymorphs graphically, because the two slopes are very similar to each other: $\delta^{13}C= 2.85*\delta^{18}O + 35.2$ (calcite) and $\delta^{13}C= 2.70*\delta^{18}O + 32.4$ (aragonite). $R^2= 0.60$ and $0.79$ for calcite and aragonite, respectively. Fig. 3 has been improved.

*Page 5, Line 6: Please replace "more negative" with "lower".*
Corrected

*Page 5, Line 8: Please replace "more positive" with "higher".*
Corrected

*Page 5, Lines 12-13: Please add that the referred Table 1 is the Supplement Table 1.*

Corrected

*Page 5, Line 29: I think you mean "Suppl. Fig. 2a" instead of "Suppl. Fig. 5".*
Yes, the reference was corrected.

*Page 6, Lines 9-10: Please give the exact temperature as a number.*
This has been added to the text.

*Page 6, Line 14: What do you exactly mean with "...primarily regarded as a proxy for $\delta^8O$ of local precipitation."? There is no amount effect in the $\delta^{18}O$ of the precipitation in the Alps. Therefore, the $\delta^{18}O$ of the precipitation should have a relation to temperature. Due to the quite long transfer time of the water you mentioned, the water should contain long-term changes in the $\delta^{18}O$ of precipitation and therefore, of long-term changes in temperature. Please add some more information to this topic at this point.*

We agree with the reviewer that there is no amount effect in the $\delta^{18}O$ data of precipitation in the Alps. We interpret the oxygen isotopes in our record primarily as a proxy of $\delta^{18}O$ in precipitation. A temperature rise of $1^oC$ would lead to 0.59±0.09‰ higher isotope values in precipitation in the mid and high latitudes (Rozanski et al., 1992). This would be partially counterbalanced by the isotope fractionation when calcite/aragonite forms. The temperature dependence of the oxygen isotope fractionation during calcite precipitation is -0.24 ‰/$^oC$ based on experimental studies (Kim and O`Neil, 1997), while a somewhat higher value (-0.18 ‰/$^oC$) was determined by a cave-based study (Tremaine et al., 2011). Kim et al., 2007 reported a similar value (-0.22 ‰/$^oC$) for the temperature coefficient for the oxygen isotope fractionation of aragonite. Consequently, for the study area a net isotope change of 0.35-0.41‰/$^oC$ and 0.37‰/$^oC$ is expected for calcite and aragonite, respectively.

Since the flowstone-forming water has a residence time of up to a decade (Spötl et al., 2002), we propose that these secondary carbonates record changes in the $\delta^{18}O$ of the precipitation and thus temperature on decadal to multi-decadal timescales.

LAS 6 is an annually laminated calcite sample. On an intra-annual time scale its $\delta^{18}O$ variability records surface temperature changes that were transmitted to the shallow subsurface by heat advection (Koltai et al., 2017). On a multi-annual time scale $\delta^{18}O$ variability of LAS 6 is replicated well by LAS 19, suggesting that the two flowstones were deposited close to isotopic equilibrium (Dorale and Liu, 2009). Thus, LAS 6 also provides insights into the changes of $\delta^{18}O$ of precipitation on (multi-)decadal timescales.

In response to the reviewer`s comment we clarify this in the revised text.

*Page 6, Line 19: Please replace "kinetic" with "disequilibrium".*
Corrected

*Page 6, Line 27: Please replace "more enriched" with "higher".*
This sentence had a mistake, which has been corrected.

*Page 6, Line 29: I am not sure if a co-variation of $\delta^{18}O$ and $\delta^{13}C$ indicate in-aquifer processes. Therefore, please delete "as indicated by the co-variation of $\delta^{18}O$ and $\delta^{13}C$ isotopes.".*

We slightly disagree with the reviewer on this. Although prior calcite/aragonite (PCP/PAP) mostly influences the carbon isotopic composition of the speleothem, laboratory studies show

that calcite precipitation results in the progressive enrichment in both $^{13}C$ and $^{18}O$ (e.g. Polag et al., 2010; Dreybrodt and Scholz, 2011). Consequently, we suggest that PCP may lead to progressively higher $\delta^{18}O$ values along the flowpath, even if this change may be much smaller in amplitude than that of $\delta^{13}C$. Laboratory studies investigating the influence of PAP on the stable isotope composition of the precipitating aragonite are lacking, but a simultaneous enrichment in $^{13}C$ and $^{18}O$ is to be expected.

In response to the reviewer`s comment, we provide further discussion on this in the text.

*Page 6, Line 35: Please replace "kinetic" with "disequilibrium".*
Corrected

*Page 7, Line 5: Please replace "kinetic" with "disequilibrium".*
Corrected

*Page 7, Line 9: Please replace "kinetic" with "disequilibrium".*
Corrected

*Page 7, Line 23: Please replace "kinetic" with "disequilibrium".*
Corrected

*Page 7, Lines 34-35: Please delete "and thus provide short snapshots of local climate.". For me a multi-decadal to centennial resolution to not provide snapshots, which are some very short time-intervals for me.*

We rewrote this sentence in order to emphasise that such fracture-lining flowstones may provide a fragmented record of the local climate history, but have only a limited potential to deliver a long (multi-millennial), continuous record.

*Page 8, Line 9: Please replace "kinetic" with "disequilibrium".*
Corrected

*Page 8, Line 13: You refer to Fig. 8, however, there is no Figure 8 in the manuscript.*
Corrected to Figs. 4 and 6.

*Page 8, Lines 16-18: Please show this in a figure. That makes it easier for the reader.*
A new supplementary figure (Suppl. Fig. 4) has been prepared.

*Page 8, Line 20: Please refer also to the specific panel in Figure 6.*
Done

*Page 8, Line 20: Please mark the Younger Dryas and the Boling-Allerod in Figure 6.*
Fig. 6 has been improved.

*Page 8: Line 30: Please replace "more negative" with "lower".*
Corrected

*Page 8, Line 35: Please replace "kinetic" with "disequilibrium".*
Corrected

*Page 9, Line 2: Please specify the in-aquifer processes.*

We do not think this is needed, because they are specified in the previous paragraph (p. 8 lines 36-37).

*Page 9, Lines 3-4: You write here about changing hydrological condition. This could be discussed a bit more in detail in the discussion. For me it did not come up so clearly from the discussion. This is more a general comment.*

The sentence has been modified for better understanding.

*Table 1: Could you perhaps mark values from aragonite and calcite in different colours? That make it straight forward for the reader.*

It would be difficult to mark the values for aragonite and calcite, because some of the samples are comprised of both polymorphs. Alternatively, we added a new column about mineralogy to make it easier for the reader.

*Figure 1: Are there samples from the vein-filling flowstones from the two other sites at Sponding and Eyrs? Do they perhaps provide time-spans missing in the other and could complete the record or give a better overlap?*

Unfortunately, there are no other flowstones samples available from the other two fractures that cover the same time period. All the flowstones covering the 14.2-10.0 ka period were part of this study with the exception of one calcite-aragonite sample (LAS 71) from SQ site. This sample was not analysed in details due to its complex growth structure.

*Figure 3: Please replace "kinetically" with "disequilibrium".*
Corrected

*Figure 4: Please replace "230Th" with "$_{230}$Th".*
Corrected

*Figure 6: Please replace "shown in (e) and (d)" with "shown in (d) and (e)".*
Corrected

*Figure 6: There is something missing in "represents the of LAS2,".*
Corrected

*Figure 6: I think the blue rectangle marks the Younger Dryas, but this is not indicated in the figure or the figure caption.*
Corrected

Yours sincerely,

Gabriella Koltai
(on behalf of all co-authors)